# A Combined Proteomics, Metabolomics and In Vivo Analysis Approach for the Characterization of Probiotics in Large-Scale Production

**DOI:** 10.3390/biom10010157

**Published:** 2020-01-18

**Authors:** Laura Bianchi, Luca Laghi, Virginia Correani, Emily Schifano, Claudia Landi, Daniela Uccelletti, Benedetta Mattei

**Affiliations:** 1Functional Proteomics Laboratory, Department of Life Sciences, University of Siena, 53100 Siena, Italy; bianchi12@unisi.it (L.B.); landi35@unisi.it (C.L.); 2Department of Agro-Food Science and Technology, University of Bologna, 40126 Cesena, Italy; l.laghi@unibo.it; 3Department of Biochemical Sciences, Sapienza University, 00185 Roma, Italy; virginia.correani@uniroma1.it; 4Department of Biology and Biotechnology “C. Darwin”, Sapienza University, 00185 Rome, Italy; emily.schifano@uniroma1.it; 5Department MESVA, University of L’Aquila, 67100 L’Aquila, Italy

**Keywords:** probiotic quality assessment, *Caenorhabditis elegans*, functional proteomics, metabolomics, oxidative stress, aging

## Abstract

The manufacturing processes of commercial probiotic strains may be affected in different ways in the attempt to optimize yield, costs, functionality, or stability, influencing gene expression, protein patterns, or metabolic output. Aim of this work is to compare different samples of a high concentration (450 billion bacteria) multispecies (8 strains) formulation produced at two different manufacturing sites, United States of America (US) and Italy (IT), by applying a combination of functional proteomics, metabolomics, and in vivo analyses. Several protein-profile differences were detected between IT- and US-made products, with *Lactobacillus paracasei*, *Streptococcus thermophilus*, and Bifidobacteria being the main affected probiotics/microorganisms. Performing proton nuclear magnetic spectroscopy (^1^H-NMR), some discrepancies in amino acid, lactate, betaine and sucrose concentrations were also reported between the two products. Finally, we investigated the health-promoting and antiaging effects of both products in the model organism *Caenorhabditis elegans*. The integration of omics platforms with in vivo analysis has emerged as a powerful tool to assess manufacturing procedures.

## 1. Introduction

Probiotics are defined by the Food and Agriculture Organization of the United Nation/World Health Organization (FAO/WHO) as “live microorganisms, which when administered in adequate amounts, confer a health benefit on the host” [1]. A number of variables are involved in the industrial production of bacterial species, i.e., growth conditions, substrates, cryoprotectants and storage conditions, all of which affect strains’ probiotic properties by influencing gene expression, protein patterns, and/or metabolic output.

Genomics is a powerful tool to identify bacterial genus/species and to detect the presence of drug resistance and virulence genes, but it cannot ascertain whether or not genetically related strains have the same probiotic properties. When fermentation conditions, growth media, and methods for production change, even genetically identical strains may show functional differences. Changes in the industrial processes have been shown to affect protein composition, chemical characteristics of the cell wall, and anti-pathogen activity of the probiotic strain *Lactobacillus rhamnosus* Lcr35 [2]. Grześkowiak studied several *L. rhamnosus* GG isolates from different probiotic products and confirmed that some beneficial properties were lost depending on the manufacturing site, notwithstanding the genetic identity of the strains tested [3]. Recently, some publications also discussed the equivalence of the VSL#3 probiotic formulation produced in two different manufacturing facilities [4,5,6].

It is a high concentration multi-strain preparation composed of eight different bacterial strains: three species of Lactobacilli, three strains of Bifidobacteria, and one species of *Streptococcus*. It has been reported, among other probiotics, to be active in the treatment of ulcerative colitis and pouchitis. When we performed our analyses, VSL#3 was produced in two facilities: one in Italy and one in the US. Although these products were commercialized under the same brand and were considered to have identical formulation, previous works have described different biological effects of the two products in terms of attenuated “clinical” signs of colitis and trans-epithelial electrical resistance, respectively [5,6].

These different biological activity may be bona fide related to changes in manufacturing conditions applied in production and storage in the two different facilities [7], that may have affected probiotic viability and functional properties by modifying gene expression and protein pattern.

The functional correlation of protein differences in complex networks is widely applied to define metabolic pathways affected in different tested conditions. Proteomics platforms may hence represent a precious tool in high-throughput profiling of probiotic formulations. Proteomic analyses has been successfully applied for the identification of proteins involved in probiotic interaction with the environment, including behavior in food matrix and ability to survive in the host intestine, as well as in detecting protein modifications that affect probiotic efficacy [8,9]. Furthermore, when comparing multiple strain formulations, the species-specificity of protein sequences allows us to assign the differential proteins to individual strains. This has a crucial relevance in analyzing probiotic dietary supplements, marketed as powder in capsules or sachets, in which the different microorganism strains cannot be separated without altering the “probiotic marketed characteristics” of the product ready for consumption.

The increasing use of probiotics for providing health benefits to the host [10] has resulted in the need for more stringent regulations for probiotic efficacy, safety, and quality. Thus, with the aim of developing an innovative and highly-performing quality-assessment approach for large scale probiotic production, we took advantage of the Italian (IT) and American (US) VSL#3 products as a case study to highlight biochemical and biomolecular differences that may occur between identical formulations produced in different sites or at different times in the same facility.

By combining functional proteomics with metabolomics investigations, we delineated different protein and metabolic profiles characterizing and distinguishing the two products. These analyses were integrated with the functional characterization of VSL#3 by in vivo experiments utilizing the nematode *Caenorhabditis elegans*. This simple multicellular organism, with a short life cycle, can be considered a powerful model to standardize probiotic manufacturing conditions in vivo. Diets based on bacteria have actually been demonstrated to play an important role in the control of nematodes lifespan [11]. Several studies on probiotic-fed nematodes have investigated functional aspects of anti-aging and innate immunity, exploring multiple genetic pathways that determine longevity [12,13,14]. Probiotic bacteria have also been reported having pro-longevity and anti-aging functions on worms [14]. In particular, many works indicate that several lactic acid bacteria (LAB) can enhance worm stress responses through different signaling pathways [11]. In this context, *C. elegans* is a valuable model system thanks to its well-characterized immune and oxidative stress responses and its highly conserved metabolism.

In conclusion, by using a combination of proteomics, metabolomics, and in vivo analyses, we obtained a multilevel characterization of IT and US VSL#3 formulations and the different characteristics of investigated formulations are discussed in relation to the production sites and expiry dates. The approach proposed in this work for probiotic functional evaluation may represent a novel tool in the quality assurance procedures of the large-scale production of probiotics.

## 2. Materials and Methods

### 2.1. Collection of the Samples of the Probiotic Products

VSL#3 is multispecies probiotic formulation produced in two different sites: Danisco/DuPont (Madison, WI, USA) and Nutrilinea/CSL (Gallarate, Italy). Commercial packaging reports the following composition: *Lactobacillus acidophilus* BA05, *Lactobacillus plantarum* BP06, *Lactobacillus paracasei* BP07, *Lactobacillus debrueckii* subsp. *bulgaricus* D08B, *Bifidobacterium breve* BB02, *Bifidobacterium longum* BL03, *Bifidobacterium infantis* BI04, and *Streptococcus thermophilus* BT01 for IT products; and *Lactobacillus acidophilus* DSM 24735, *Lactobacillus plantarum* DSM 24730, *Lactobacillus paracasei* DSM 24733, *Lactobacillus debrueckii* subsp. *bulgaricus* DSM 24734, *Bifidobacterium longum* DSM 24736, *Bifidobacterium breve* DSM 24732, *Bifidobacterium infantis* DSM 24737, and *Streptococcus thermophilus* DSM 24,731 for US products.

Despite that VSL#3 strains were reported with different names in the two products and the relative concentrations of the eight strains components were not given, Italian and American products were commercialized under the same brand and the producer did not declare any difference in composition between them. Hence, as in previous works [4,5,6], we considered the two products as having identical formulation.

Ten distinct lots of VSL#3 produced in different sites and having diverse expiry dates (as listed in Appendix A) were used for proteomic analysis: seven lots for metabolomics analysis and two lots on the *Caenorhabditis elegans* animal model. Batches were maintained according to the manufacturer’s instructions until use.

### 2.2. Mass Spectrometry

For proteomic analysis, 20 mg of each VSL#3 capsule batch were separately resuspended in 300 µL of lysis buffer (8 M urea, 30 mM HEPES, 1 mM phenylmethylsulfonyl fluoride) containing complete protease cocktail inhibitors (Roche, Mannheim, Germany).

Samples were vigorously vortexed, sonicated on ice (5 cycles of 20 s each with 30 s of cooling between each on ice), and centrifuged at 16.800× *g* for 30 min at 4 °C. Protein quantification of supernatants was performed by Bradford assay.

Each sample (~50 µg) was reduced, alkylated, and trypsin digested, as described in [15]. Resulting tryptic peptides were fractionated by strong cationic exchange (SCX) into two fractions (50 and 500 mM ammonium bicarbonate), which were then desalted by Stage tip protocol using C18 reversed-phase resin, speed-vac dried and finally resuspended in 60 µL 0.1% (*v/v*) formic acid for mass spectrometry analysis.

Nano liquid chromatography tandem mass spectrometry (LC-MS/MS) analyses were performed on a quadrupole-Orbitrap mass spectrometer (Q-Exactive Plus; Thermo Fisher, Waltham, MA, USA) coupled with a nano-Ultra Performance Liquid Chromatography (nano-UPLC) system (Thermo Fisher) equipped with a reversed phase C18 nanocapillary analytical column (75 μm i.d. × 15 cm) (Thermo). Three technical replicates were run for each fraction.

Tryptic peptides were separated using a 100 min linear gradient formed by solvent A (0.1% (*v/v*) formic acid in water) and solvent B (0.1% (*v/v*) formic acid in acetonitrile). Afterwards, they underwent electrospray at 1.8 kV and their tandem mass spectra from the top 20 ions, in a range of 350–2000 *m/z* values, were acquired applying conventional parameters [16].

### 2.3. Proteomic Data Analysis

MS data were analyzed by means of MaxQuant software (v. 1.6.0.1, Max Planck Institute of Biochemistry, Planegg, Germany). For protein identification, a custom database was built combining the UniProtKB reference proteomes (release 4 February 2017) for *Lactobacillus acidophilus* (strain ATCC 700396/NCK56/N2/NCFM, 1859 protein counts), *Lactobacillus helveticus* (strain DPC 4571, 1580 protein-counts), *Lactobacillus plantarum* (strain ATCC BAA-793/NCIMB 8826/WCFS1, 3087 protein-counts), *Lactobacillus paracasei* (strain ATCC 334/BCRC 17002/CIP 107868/KCTC 3260/NRRL B-441, 2708 protein-counts), *Bifidobacterium breve* (strain DSM 20213, 2262 protein-counts), *Bifidobacterium longum* (strain NCC 2705, 1725 protein-counts), *Bifidobacterium infantis* (strain ATCC 15697/DSM 20088/JCM 1222/NCTC 11817/S12, 2399 protein-counts), and *Streptococcus thermophilus* (strain ATCC BAA-250/LMG 18311, 1577 protein-counts; strain CNCM I-1630, 1889 protein-counts). According to its reclassification, we reported *Lactobacillus delbrueckii* subsp. *bulgaricus* as *Lactobacillus Helveticus* in our database. After their search against common contaminant and reversed decoy databases, MS/MS spectra were analyzed selecting the following parameters: two allowed missed-cleavages for trypsin, methionine oxidation and protein N-terminal acetylation as variable modifications, carbamidomethylation of cysteine as fixed modification. False Discovery Rate (FDR) of the identification was set to 1% at protein and peptide levels. The “Match between runs” feature (2 min time window) allowed identification transfer across experiments to minimize missing values.

Protein quantifications were based on the identified “razor + unique peptides”. The intensity values were also normalized by Label-Free Quantification (LFQ) algorithm to account for potential differences in sample loading. The derived LFQ Intensity was used for further proteome comparison.

Statistical analyses were performed with Perseus software (v. 1.6, Max Planck Institute of Biochemistry). Potential incorrect identifications and contaminants (identified in “Only identified by site”, “Reverse”, and “Potential contaminant” columns) were filtered out. Protein intensities were log2-transformed. Data were also filtered to have “3” valid values in at least one sample group. Missing values (protein not identified in a run) were imputed by creating a normal distribution of random numbers with a width of 0.3 relative to the standard deviation of the measured values and 1.8 standard deviation down-shift of the mean, simulating the distribution of low signal values. Data normalization occurred by subtracting the median.

Significant differences in protein abundance were assessed by Student’s *t*-test, using permutation-based FDR of 5% to control for multiple hypothesis testing one-way and an S0 value of 2.

Hierarchical clustering of proteins was performed after z-score normalization of the data, using Euclidean distances. Principal components analysis (PCA) was obtained, without z-scoring, by using two-dimensional non-parametric Mann–Whitney test, with a Benjamini–Hochberg multiple hypothesis testing correction using an FDR threshold of 0.05.

Gene ontology (biological processes) and Kyoto Encyclopedia of Genes and Genomes (KEGG) annotations, describing differentially abundant proteins, were retrieved from UniProtKB. In order to test whether some GO terms were over-represented among the differentially abundant proteins, the DAVID (the database for annotation, visualization and integrated discovery) bioinformatic resource was used to identify enriched terms [17].

Further GO and KEGG-pathway annotation enrichment analyses were performed applying STRING (Search Tool for the analysis of INteracting Genes/Proteins). This resource was also used to obtain protein–protein interaction enrichments based on published or informatics-predicted functional interactions [18]. The evidence annotation in STRING was filtered out of interactions from text-mining and neighborhood, and only interactions supported by experimental evidence, co-expression, and existing database information with high-confidence score >0.7 were considered. Protein–protein interaction and GO/KEGG-term enrichments were retained significant with an FDR *p* < 0.001 and *p* < 0.005, respectively.

The STRING database does not support *L. paracasei* and *B. infantis* as reference organisms. Based on their phylogeny, *L. paracasei* and *B. infantis* identifications were consequently processed, selecting *L. casei* and *B. longum* as reference organisms, respectively [19,20].

### 2.4. Metabolomic Analysis

The probiotic products were dissolved at 1% (*w/v*) in 0.9% NaCl solution. After centrifugation at 5000× *g* for 10 min, supernatants were filtered through a 0.2 μm membrane filter and stored at −20 °C. Filtered supernatants derived from 0.9% (*w/v*) NaCl suspension and MRS (Man Rogosa Sharpe agar) culture of the VSL#3 products were prepared for NMR analysis by thawing and centrifuging them for 15 min at 18,630× *g* and 4 °C. The supernatants, 0.7 mL, were added to 0.1 mL of a D_2_O solution of 3-(trimethylsilyl)-propionic-2,2,3,3-d4 acid sodium salt (TSP) 10 mM, kept at pH 7.00 ± 0.02 with 1 M phosphate buffer and centrifuged again.

An AVANCE III spectrometer (Bruker, Milan, Italy), operating at a frequency of 600.13 MHz, was employed to record ^1^H-NMR spectra at 298 K. The HOD residual signal was suppressed by means of presaturation. Following the procedure described in [21], the signals from broad resonances originating from large molecules were suppressed by a CPMG-filter, while the HOD residual signal was suppressed by means of presaturation. Each spectrum was acquired by summing up 256 transients using 32 K data points over a 7184 Hz (for an acquisition time of 2.28 s). NMR analysis was carried out as previously reported in [22]. Briefly, the recycle delay was set to 5 s, keeping into consideration the longitudinal relaxation time of the protons under investigation. The signals were assigned by comparing their chemical shift and multiplicity with Chenomx software data bank (Chenomx Inc., v. 8.3, Edmonton, AB, Canada).

### 2.5. NMR Data Analysis and Statistics

After ^1^H-NMR spectra pre-processing, as described in [22], molecule quantification was performed in the VSL#3 products by considering TSP as the internal standard in sample US-5e. The other samples were normalized towards it by probabilistic quotient normalization, excluding the signals from trehalose [23]. The concentrations were expressed as mmol/gram of product. Molecule quantification was performed in a pure MRS sample by employing TSP as the internal standard. The other samples inoculated in MRS were normalized to pure MRS, as described above. The concentrations were expressed as mmol/L.

The trends underlying metabolomics data were highlighted by robust principal component analysis (rPCA) on the centered and scaled molecules concentrations, according to [24]. This was done with the corresponding function in the rrcov package for R computational language, by considering an alpha value of 0.9 [25]

### 2.6. C. elegans Strains and Growth Conditions

The wild-type *C. elegans* strain N2 and the CF1553 (muls84[pAD76(Sod-3::GFP)]) transgenic strain used in experiments were propagated on nematode growth medium (NGM). Worms were fed with US-7 or IT-3 sample. Each product, suspended in sterile H2Odd, corresponding to 10 mg of bacterial cells in 25 μL, was spread on 3.5 cm diameter NGM plates, as described in [9].

### 2.7. C. elegans Lifespan Assay

To obtained a synchronized worm population, N2 adults were placed to lay embryos for 8 h on NGM plates containing a lawn of US-7, IT-3 samples, or *E. coli* OP50, and then sacrificed. All lifespan assays started when the progeny became fertile (t0). Animals were transferred to new plates seeded with fresh lawns and monitored daily. They were scored as dead when they no longer responded to gentle touch with a platinum wire. At least 60 nematodes per condition were used in each experiment.

### 2.8. Aging Markers

Pharyngeal pumping, locomotion behavior and lipofuscin accumulation were analyzed as described in [26,27]. The analysis was performed on 13-day adult worms, grown on US-7, IT-3, or *E. coli* OP50 starting from embryo stage. About 15 worms for each experimental condition were monitored. Lipofuscin autofluorescence was detected by fluorescence microscopy (Zeiss Axiovert 25, Carl Zeiss AB, Sweden) using a GFP filter set.

### 2.9. Oxidative Stress Analysis in C. elegans

ROS formation in *C. elegans* was measured using the fluorescent probe 2,7dichlorodihydrofluorescein diacetate (H2DCFDA, SIGMA-ALDRICH), with minor modifications [28]. Briefly, N2 synchronized adult worms were exposed to US-7 or IT-3 from embryo hatching and collected at the stage of 2-day adults. For each experimental condition, 15 worms, in triplicate, were monitored. The measurement was performed using a multiplate reader (Promega, GloMax multidetection system, Madison, WI, USA) at excitation/emission wavelengths of 485 and 520 nm.

### 2.10. Analysis of C. elegans Strain SOD-3::GFP Fluorescence

Synchronized transgenic worms were transferred daily on NGM plates containing different products and incubated at 16 °C, as described above. At the stage of 2-day adults, the fluorescence was analyzed [29]. The experiments were repeated in triplicate and 15 worms for each condition were used in each experiment. The fluorescence was observed under a Zeiss Axiovert 25 microscope.

### 2.11. Statistical Analysis of In Vivo Experiments in C. elegans

All experiments were performed at least in triplicate, and data are presented as mean ± standard deviation (SD). The statistical significance was determined by Student’s *t*-test or one-way ANOVA analysis coupled with a Bonferroni post-test (GraphPad Prism 5.0 software, GraphPad Software Inc., La Jolla, CA, USA). Significant differences were indicated as follows: ** *p* < 0.01, and *** *p* < 0.001.

## 3. Results

### 3.1. Protein Profiles of IT- and US-Made Products

To determine a quantitative proteomic profile of bacterial strains under investigation, we prepared protein extracts from 10 distinct lots of VSL#3 produced in Italy (*n* = 3) or in United States (*n* = 7:3 expired (USe) and 4 not expired (US)) and having diverse expiring dates (Appendix A). As the VSL#3 production only recently moved to Italy, expired IT-prods were not available when the analyses here described were performed.

A total of 3685 proteins at a false discovery rate of 1% were identified by nano LC-MS/MS analysis; among these, 2553 proteins could be consistently quantified in all the replicates of at least one sample (Appendix A). Quantitative differences occurring among proteomic profiles of different lots were detected by comparing their protein intensities with those from the US-4 product that was used as control group, being the most recent lot produced at Danisco/Dupont. Protein abundance differences (fold change > 1.5 or <−1.5 and *p* ≤ 0.05) principally occurred between US-4 and the IT-1, IT-2, IT-3, and US-3e samples, as summarized in Appendix A and visualized in Figure 1.

Based on the variance among depicted protein patterns, principal component analysis highlighted similarity correlations existing among analyzed samples. These grouped into two distant clusters in the PCA plot, as shown in Figure 2A. The main cluster included all the US-made products, while the other cluster grouped only IT-1 and IT-2 samples. In the plot, the IT-3 sample localized independently and consistently distant from the other Italian lots.

In agreement with PCA analysis, the unsupervised hierarchical clustering of protein abundance showed that US-4 was closely related to US-2 and US-6/7, and clearly separated from IT-1, IT-2, and IT-3 samples (Figure 2B).

The comparison of US-lots pre- and post-expiry dates was performed to evaluate the effect of time on proteomic profile and related probiotic qualities of the formulation. As highlighted by PCA and hierarchical clustering (Figure 2A,B, respectively), all US products closely cluster regardless of expiry. This similarity was observed despite the fact that expiration dates of those products spanned over 4.5 years.

Subsequent results (presented below) involve comparisons between the not-expired US- and IT-products. According to the generated in-house protein database, sequence information on peptides obtained by mass spectrometry allowed the differential identification of bacteria species present in the sample. About two-thirds of quantified proteins were found to significantly differ in abundance between the US and the IT-made groups. While 848 proteins were more abundant in US samples, 696 proteins increased in IT lots. Based on identification score prioritization, the overwhelming majority of differences were contributed by *L. paracasei*, *S. thermophilus*, and *B. breve* species. Roughly 39% and 35% of proteins more abundant in US-made lots were from *S. thermophilus* and *B. breve*, respectively (Figure 3A).

According to *B. infantis* reclassification as a subspecies of the *B. longum* species, differences detected in *B. longum* and *B. infantis* were considered as identifications from the same *Bifidobacterium* species [19,30,31]. *B. longum* hence contributed to about 10% of protein differences with increased abundance in the US-products. Interestingly, slightly less than half (45%) of proteins over-present in the US-products was identified from Bifidobacteria. On the other hand, 81% of proteins more abundant in Italian products was identified from *L. paracasei* (Figure 3B).

To investigate the biological and functional meaning of the highlighted differences, we performed unsupervised hierarchical clustering of Gene Ontology (GO) Biological Process (BP) terms and KEGG pathway annotations that describe the abundance of differing proteins. For this purpose, the DAVID bioinformatic resource was applied and functional enrichments were independently calculated for the two groups of differential proteins: (i) proteins more abundant in US-lots and (ii) proteins enriched in IT-lots. Groups (i) and (ii) were processed regardless of identifying species.

In addition to generic terms (e.g., “Biosynthesis of secondary metabolites” (lpl01110), “Microbial metabolism in diverse environments” (stl01120), and “Metabolic pathways” (lpl01100)), US products showed significant enrichment of GO and KEGG pathway terms annotating protein factors involved in specific processes, i.e., “biosynthesis of antibiotics” (lpl01130, stl01130), “biosynthesis of amino acids” (lpl01230), “cell redox homeostasis” (GO: 0045454), “ATP-binding cassette (ABC) transporter complex” (GO: 0043190), and “glycolysis/gluconeogenesis” (lpl00010) (Figure 4A).

As underlined by KEGG pathway ID, where “lpl” means *L. plantarum* and “stl” *S. thermophilus* (Figure 4A), the significantly enriched terms were mainly related to *L. plantarum* and *S. thermophilus*. Despite the low percentage (6.5%) of *L. plantarum* proteins more abundant in US-made lots, several significantly enriched KEGG pathways were found related to *L. plantarum* identified differences. This suggests the probable occurrence of a significant differential behavior of the bacterium in US-products compared to the corresponding strain in IT samples.

Among the proteins more abundant in the IT products, almost all of the enriched KEGG pathway annotations belonged to *L. acidophilus*. Enriched processes included synthesis of peptidoglycan (lca00550), aminoacil-tRNA (lca00970), and antibiotics (lca01130); DNA repair (lca03430); glycolysis and gluconeogenesis (lca00010); fructose and mannose metabolism (lca00051); and ribosome activity (lca03010) (Figure 4B).

Despite the high number of differentially abundant proteins, Bifidobacteria and *L. paracasei* did not apparently present specific metabolic-pathways that may differentially operate in the two types of probiotics.

### 3.2. STRING Analysis

Protein accession numbers of several significant differences were not recognized by DAVID, thus resulting in their exclusion from the analysis. To better characterize the biological terms describing the differentially abundant proteins, identified proteins (more abundant in IT- or US-made formulations) were further analyzed, species by species, using the bioinformatic resource STRING [18]. STRING is commonly used to build protein–protein interaction networks on a global scale based on de novo predicted associations, GO annotations, and KEGG pathways [32]. The software allows processing a multiple sequence list against the entire protein dataset of a specific selected organism that is included in the STRING database. In addition, sequence similarity search and homology-based information annotated by the program improve enrichment results by increasing annotations and the number of recognized protein sequences [32]. Hence, GO and KEGG-pathway enrichments obtained with STRING may differ from those obtained using DAVID.

Here, STRING highlighted significant enrichment of numerous GO terms and/or KEGG pathways from *L. (para)casei* and *B. breve* proteins that were more abundant in IT and US products, respectively (Appendix A). Highly significant term enrichments were also identified among proteins over-present in *S. thermophilus* from US lots (Appendix A).

Obtained term enrichments suggested an increase of proteins involved in cellular homeostasis and proliferation. These pathways spanned from DNA synthesis and repair to gene expression, from amino acid synthesis to protein metabolism, from carbohydrate metabolic processes to ATP synthesis, and from anabolic processes to cell division. According to GO Cellular Component enriched terms, the delineated protein-increase affected almost all cellular compartments. *L. (para)casei* presented the highest number of significant enrichments as well as the highest values of statistical significance (Appendix A). Noteworthy, proteins involved in protein synthesis, transmembrane trafficking, and protein folding were enriched in US *L. plantarum*, *S. thermophilus*, and bifidobacteria as well as in IT *L. (para)casei*. In particular, the increment of the main chaperon systems active in heat stress response of probiotic lactobacilli and bifidobacteria [7] (i.e., the GroEL chaperonin (in US *L. plantarum* and IT *L. paracasei*), the molecular chaperone DnaK (in US *L. plantarum* and bifidobacteria and in IT *L. paracasei*), the heat shock protein GrpE (in *B. longum*), as well as members of the Clp machinery (in US *B. breve* and *S. Thermophilus* and in IT *L. paracasei*) and the proteases FtsH (in US *S. Thermophilus* and IT *L. paracasei*) and HtrA (in US *S. Thermophilus*)) (Appendix A).

While the US *S. thermophilus* protein–protein interaction enrichment (PPIe) was, even if slightly, not significant (PPIe *p*-value = 0.0014), highly-significant and complex PPI enrichments were highlighted by STRING processing of Italian *L. (para)casei* (PPIe *p* < 1 × 10^−16^) and American *B. breve* (PPIe *p* < 1 × 10^−16^) abundance differing proteins. Beside *S. thermophilus*, DAVID analysis highlighted *L. plantarum* and *L. acidophilus*, presenting the main significant GO and KEGG pathway annotation enrichments; Figure 5A,B, respectively. STRING evidenced a highly significant PPIe (*p* < 1.6 × 10^−16^) and significant GO and KEGG term enrichments by processing *L. plantarum* increased abundance proteins in US-made products (Appendix A). Conversely, *L. acidophilus* more abundant proteins in Italy-made products did not present significant STRING PPIe (PPIe *p* = 0.0044). They also present poorly significant GO and KEGG pathway enrichments (Appendix A).

Based on the *B. infantis* reclassification, as above detailed, *B. longum* and *B. infantis* identified differences with increased abundance in US-products were co-processed by STRING setting *B. longum* as reference species. This led to a number of significant GO and KEGG pathway term enrichments and a highly significant PPIe value (= 1.09 × 10^−10^), as shown in Appendix A.

Based on the proteomic analysis, *L. paracasei*, *S. thermophilus*, and Bifidobacteria accounted for much of the quantitative differences detected between IT and US-made products. The bioinformatic analyses performed by DAVID and STRING on differential proteins provided a general functional-overview about pathways probably affected by different manufacturing facilities. The proteins with increased abundance in IT *L. (para)casei* and in US Bifidobacteria and *S. thermophilus* were essentially involved in basic metabolism. Due to their high hierarchy levels, the GO and KEGG pathway annotation enriched terms did not reveal critical differences in specific pathways of bacterial metabolism.

### 3.3. Metabolomic Analysis

In VSL#3 products, we quantified by ^1^H-NMR 35 molecules, listed in Appendix A, mainly pertaining to the classes of amino acids and organic acids. In order to highlight similarities among the samples, we calculated an rPCA model over their concentrations (Figure 5). Two components were found to robustly represent the data. Along principal component PC1, accounting for the 74.3% of the described variance, the samples IT-1 and IT-2 appeared at the highest scores, while US-made samples appeared at the other extreme of the PC. The correlation plot between molecules concentration and their importance over PC1 (Figure 5B) provides an overall impression of the specific molecules leading to this grouping.

Samples produced in Italy were characterized by higher concentrations of tryptophan, phenylalanine, tyrosine, isoleucine, leucine, valine, lysine, and lactate, while US-samples had higher concentrations of sucrose, glycine, arginine, and betaine.

PC 2, accounting for the remaining 25.7% of the explained variance, was mainly influenced by the expiry date. US samples analyzed before expiry tended to be characterized by higher concentrations of 4-aminobutyrate, isobutyrate, acetate propylene glycol, and glutamate, while those analyzed after expiry date tended to show higher concentrations of glucose.

### 3.4. Impact of the Formulations on C. elegans Physiology

US-7 and IT-3 samples were evaluated in *C. elegans*, utilized as a model organism to screen for probiotic bacteria [14,33,34]. The first phenotype analyzed was the lifespan of US-7 or IT-3 fed animals, starting from embryo hatching, compared with worms fed the standard diet based on *Escherichia coli* OP50, taken as controls. Results show that both products induced a significant increase in nematode longevity. In particular, nematode median survival was recorded at days 16 and 17 when worms were fed with IT-3 and US-7, respectively, as compared to day 11 in the case of OP50-fed worms (Appendix A).

In order to evaluate effects on *C. elegans* aging, pharyngeal pumping, body movement, and lipofuscin accumulation were analyzed in 13-day adult nematodes supplemented with US or IT formulations. Neuromuscular functionality, examined through the measure of pumping rate, increased in both US-7 and IT-3 fed nematodes, with respect to the OP50 control (Appendix A). Body bending is also related to age, becoming weaker in old worms. Measuring the locomotory rate of *C. elegans* on day 13, worms fed the two products displayed a higher body movement than OP50-fed worms (Appendix A). Lipofuscin accumulation is a biomarker of ageing in *C. elegans*. Fluorescence microscope analysis revealed, in IT-product fed worms, an increase in autofluorescent lipofuscin granules, diffusing throughout the nematode gut. By contrast, nematodes fed with US-7 showed a reduced fluorescence signal, similar to OP50-fed worms (Figure 6A).

As the *C. elegans* in vivo model permits us to investigate the effects of a bacterial diet also at the cellular/molecular levels, we decided to perform further experiments to ascertain the influence of the two products on stress resistance, since extended longevity and enhanced stress resistance are correlated [35,36]. Superoxide Dismutase 3 (SOD-3) is an iron/manganese superoxide dismutase involved in the defense against oxidative stress and promoting normal lifespan [37]. The effect of feeding nematodes with American and Italian formulations was therefore analyzed in a *C. elegans* transgenic SOD-3::GFP strain. Microscopy analysis showed a fluorescence intensity higher than 10% in US-7 fed worms, as compared to control (Figure 6B,C), while IT-3 fed worms showed a fluorescence reduction of about 20%, with respect to OP50. Finally, to detect eventual differences in Reactive Oxygen Species (ROS) accumulation between worms grown on different bacteria lawns, intracellular ROS levels were investigated by using H2DCF-DA as molecular probe. This molecule is a membrane-permeable non-fluorescent probe, which can enter worm cells where it is converted to H2DCFs. This probe is then oxidized by ROS to yield the fluorescent dye DCF. Figure 6E shows 2-day adult worms fed with US-7 present a 90% reduction of fluorescence intensity when compared with OP50 fed worms, and a 30% reduction as compared to IT-3.

## 4. Discussion

In this study, we report that the site of manufacture apparently represents the driver of proteomics and metabolomics differences detected among the different analyzed VSL#3 products, and that those differences can affect a model organism consuming VSL#3.

Our investigation revealed many quantifiable differences in protein abundance between IT and US products as well as species-specific differential distribution of differential proteins. In particular, *L. paracasei*, Bifidobacteria, and *S. thermophilus* were mainly affected among Italy and US-made not-expired products. DAVID and STRING processing of identified differential proteins provided a general functional overview of pathways that were affected in those bacteria. Bioinformatic analyses indicated that proteins increased in abundance in IT *L. (para)casei*, US Bifidobacteria and, at least in part, US *S. thermophilus* were essentially involved in basic metabolic processes. According to their high hierarchy levels, GO term and KEGG pathway annotation enrichment analyses did not reveal, in the mentioned species, differential occurrence of specific metabolic pathways between the investigated probiotic groups. Rather, they suggested a generalized abundance shift of metabolic proteins between US-made and Italy-made products that may be related, among others, to fluctuations in relative concentration of individual strains or to their metabolic activity.

Worthy of note, GO-term enrichment analysis indicated significant enrichment of more specific metabolic pathways in *L. plantarum* and *S. thermophilus* from the US product, i.e., (i) the biosynthesis of antibiotics and amino acids, (ii) cell redox homeostasis, (iii) ATP-binding cassette (ABC) transporter complex, and (iv) glycolysis/gluconeogenesis (Figure 4A). Similarly, *L. acidophilus* from the IT product presented enrichments of specific KEGG pathway annotations, i.e., (i) synthesis of peptidoglycan, aminoacil-tRNA, and antibiotics, (ii) DNA repair, (iii) glycolysis and gluconeogenesis, (iv) fructose and mannose metabolism, and (v) ribosome activity (Figure 4B). These results underline possible increased activity in metabolic pathways that may have implications in microorganism pre-adaptation and cross-protection phenomena.

Microorganisms encounter stress conditions that can occur during processing and storage and can affect viability. The impact of exposure to stress conditions on the functional properties of surviving probiotic microorganisms may have important relationships with probiotic functional properties and robustness, such as resistance to osmotic, oxidative, heat and cold stresses, and tolerance to antibiotics [7].

Also, STRING analysis of differential proteins showed significant functional integration of proteins involved in protein synthesis, transmembrane trafficking, and protein folding in US *L. plantarum*, *S. thermophilus* and Bifidobacteria as well as in IT *L. (para)casei*; in particular, the increment of the main chaperon systems active in heat stress response of probiotic lactobacilli and Bifidobacteria. These results may indicate differential occurrence of heat stress during technological processing between individual strains of the two products. Conversely, the increase of ribosomal and membrane proteins in the same strains is indicative of microorganism vitality, allowing us to hypothesize that heat stress, if occurred, had not compromised the functionality of membrane systems and protein synthesis, which would otherwise have resulted in cell death [7].

A metabolomic approach was undertaken to determine differences in metabolite occurrence and abundance among IT and US-made products, and possibly evaluate the correlation with the differential metabolic pathways. ^1^H-NMR detected significant differences in amino acid abundance among analyzed lots. Italy-made probiotics showed the highest levels of seven amino acids, while US-made ones presented only arginine and glycine increase. Of course, it is not possible to unambiguously ascribe a discrepancy in amino acid content to differences in amino acid metabolism or composition of fermentation medium between the two products. Nonetheless, STRING analysis of proteins more abundant in US-prods showed the KEGG pathway annotation “biosynthesis of amino acid” (map01230) significantly enriched in both *S. thermophilus* (FDR = 4.29 × 10^−5^) and *B. breve* (FDR = 0.0003) strains. The former also presented significant enrichment of the GO term “cellular amino acid biosynthetic process” (GO: 0008652; FDR = 0.004). No specific-term significant enrichments were instead observed in IT-made products, suggesting an increment of amino acid biosynthesis.

As bacteria do not synthesize it, the higher sucrose concentration observed in US lots was likely due to differences in growth medium composition. The high content of lactate supports the hypothesis that *L. paracasei* is present in higher amounts in Italian products or that it has a high fermentative activity. *L. paracasei* is a facultative heterofermenter that, when maintained in proper growing conditions, prevalently ferments monosaccharides to lactic acid [20].

Finally, the increase in betaine concentration observed in US-made lots, even if due to the manufacturer’s supplementation, implies that the US food supplement may play a role for host detoxification of homocysteine. Indeed, VSL#3 has been shown to reduce homocysteine in clinically relevant levels among older adults [38]. Given that homocysteine is a risk factor for CAD (coronary artery disease), this difference between products could be relevant [39].

In this work, the health-promoting and antiaging effects of different manufactured probiotics were assessed on the model organism *C. elegans*. Lifespan experiments with wild type nematodes highlighted pro-longevity properties of both US-7 and IT-3. An increased pumping rate and body-bending correlates with a delayed probiotic-induced aging process and with prolongation of animal lifespan [40,41]. This finding is not surprising since it has been reported that other probiotics, i.e., *Lactobacillus plantarum* K90 and *L. paracasei* CD4, showed similar effects when administrated in diet to *C. elegans* [42]. Also, Bifidobacteria contribute to many beneficial effects on nematode physiology, longevity, and oxidative stress responses [12,41,43]. However, further experiments showed that the oxidative stress response was different, depending on the formulation administered to the nematodes. These data are in agreement with the DAVID analysis showing a significant enrichment of cell redox-homeostasis related protein in the US VSL#3 product.

It has been demonstrated that the aging process is correlated to increased oxidative stress and that lipofuscin reports oxidative damage [44]. During aging, the lipofuscin biomarker accumulates in *C. elegans* gut because of oxidative degeneration [45]. Transgenic SOD-3::GFP worms, when fed US-7, exhibited a higher fluorescence of the reporter protein as compared to *C. elegans* fed IT-3. In addition, US-7-fed nematodes showed reduced levels of ROS with respect to worms fed IT-3.

High levels of ROS induce loss of protein, lipid, and DNA functions, leading to inflammation, cytotoxicity and, in extreme cases, to cell death. Nonetheless, ROS are also known to contribute to anti-senescence [46]. For assessing probiotic functionality, lifespan experiments with wild type nematodes could hence be misleading if not accompanied by an assessment of the ROS amount and expression of the oxidative stress-related genes such as *sod-3*.

## 5. Conclusions

Living organisms express genes and modulate their expression levels in relation to environmental characteristics and changes. In probiotic-formulation production, dissimilarities in manufacturing protocols and procedures may indeed determine differences in protein patterns and metabolic networks even among identical probiotic bacteria strains. Different protein and metabolic profiles may result in different probiotic vitality, robustness, and functionality. As a rule, the development of an effective analytical method is urgent for qualitative assessment of probiotic products.

Toward this goal, we propose an integrated multiple-level molecular-analysis to investigate the consistency in the manufacturing of a formulation. By combining proteomics, metabolomics, and in vivo experiments, we conclude that even slight modulation in industry large-scale production of probiotic microorganisms may result in a wide affection of their metabolic pattern. Here, we presented and discussed some of those affections, supposedly according to bioinformatics predictions and partially proved by in vivo analyses.

## Figures and Tables

**Figure 1 biomolecules-10-00157-f001:**
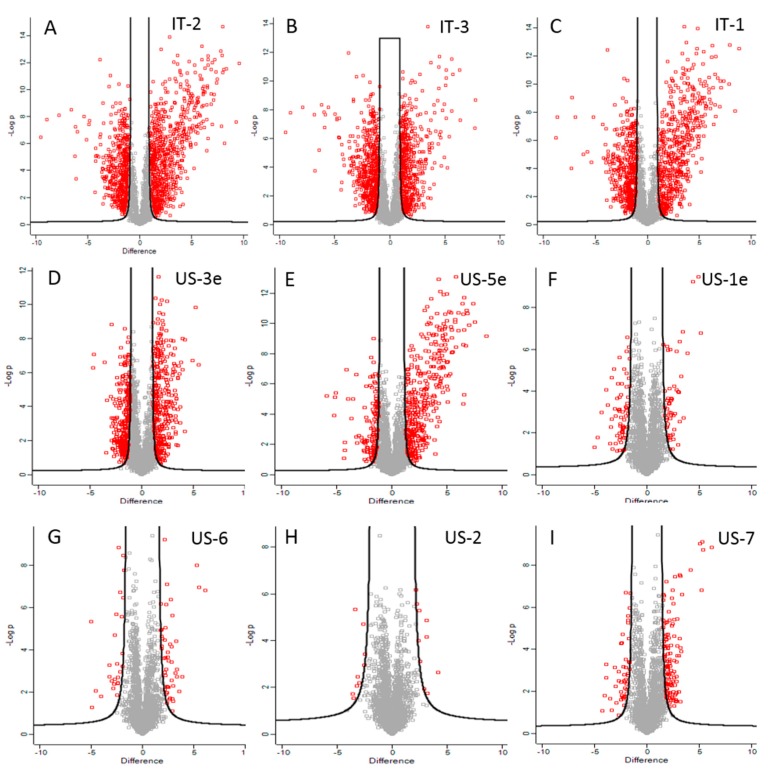
Perseus volcano plots representing proteins with differential abundance (red squares) between the tested VSL#3 samples and US-4. For each protein, significance expressed as *p*-value was graphed in function of difference between samples (log2 fold change). Red squares in the right area of each volcano plot represented proteins significantly more abundant in IT- and US-products while red squares in the left side of plots corresponded to proteins more abundant in the US-4 product.

**Figure 2 biomolecules-10-00157-f002:**
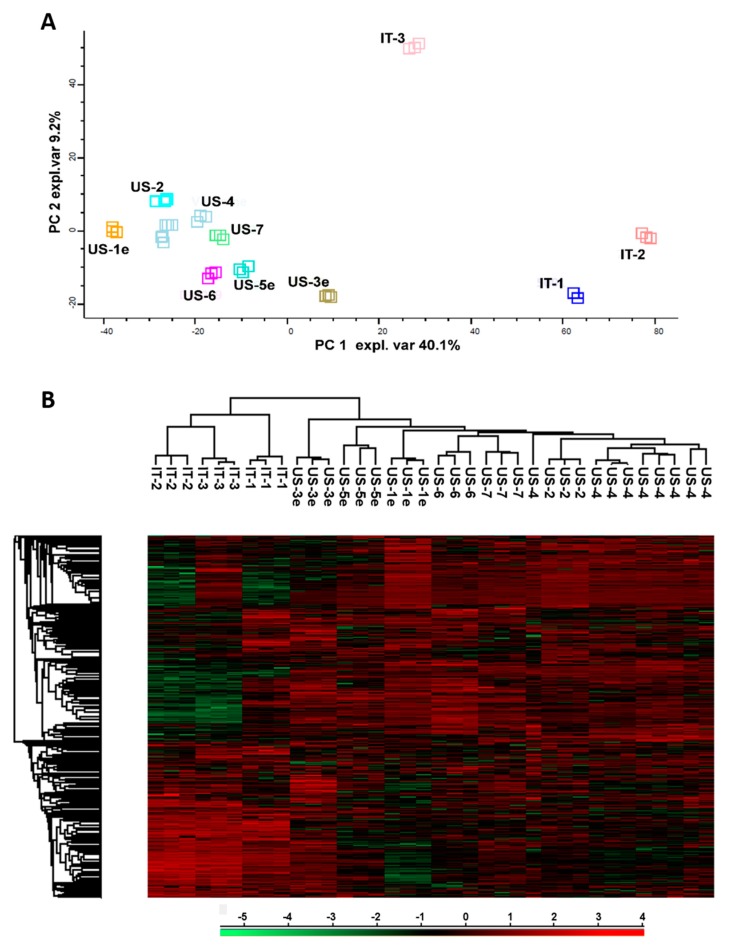
(**A**) Principal component analysis of quantified proteins highlighted in VSL#3 products. (**B**) Unsupervised hierarchical clustering of proteins identified in VSL#3 products by LC-MS/MS performed with Perseus software.

**Figure 3 biomolecules-10-00157-f003:**
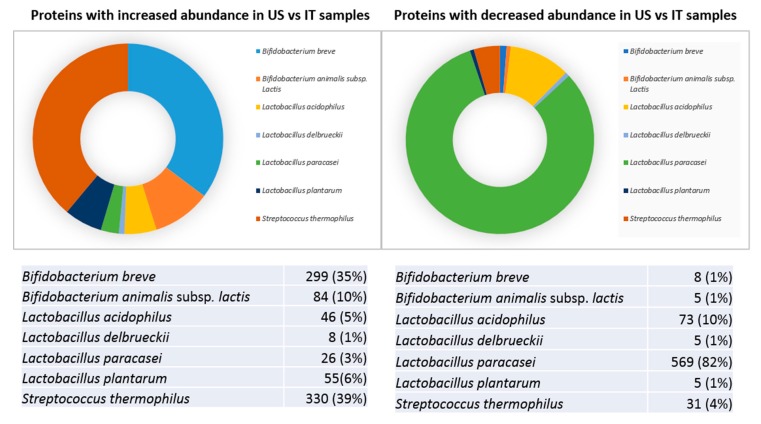
Proteomic representation of different strains in proteins with increased (**A**) or decreased (**B**) abundance in US vs. IT samples.

**Figure 4 biomolecules-10-00157-f004:**
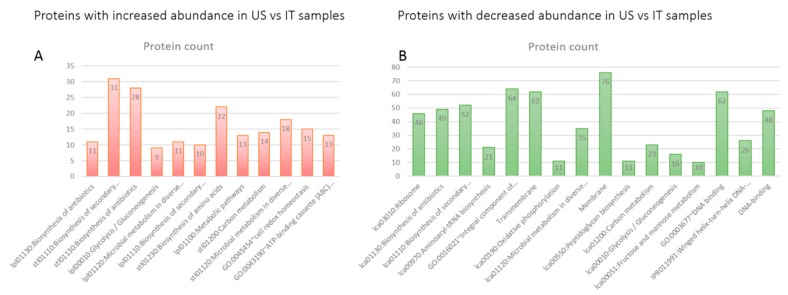
Database for Annotation, Visualization, and Integrated Discovery (DAVID) enrichment analysis of proteins with increased abundance in US-(**A**) and IT-products (**B**).

**Figure 5 biomolecules-10-00157-f005:**
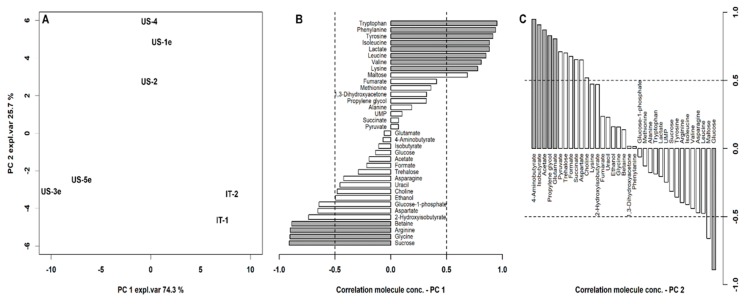
(**A**) Score plot of an rPCA model on the molecules quantified by proton nuclear magnetic spectroscopy (^1^H-NMR) on VSL#3 products. (**B**,**C**) Correlation between molecules concentration and their importance over PC1 and 2, respectively. Gray bars highlight statistically significant correlations (*p* < 0.05).

**Figure 6 biomolecules-10-00157-f006:**
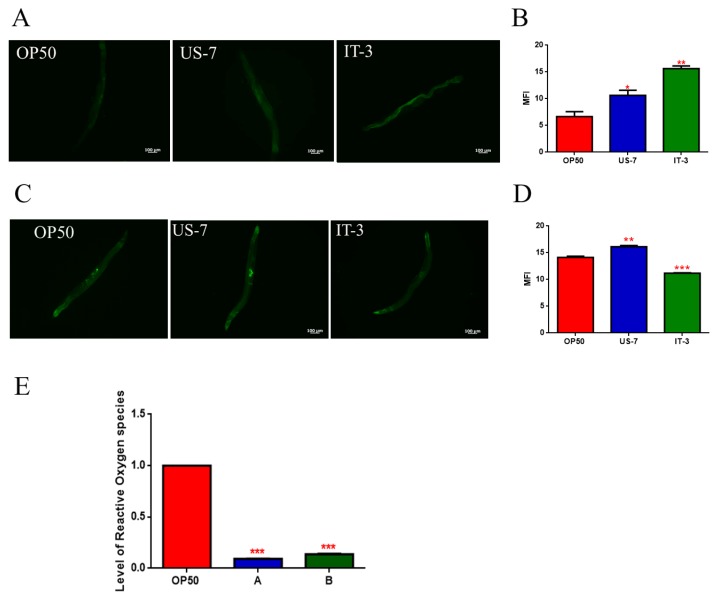
Lipofuscin and oxidative stress analysis in nematodes fed OP50, US-7, or IT-3. (**A**) Autofluorescence of lipofuscin granules in *C. elegans* wild type strain fed OP50, US-7, or IT-3 on day 13. Scale bar = 100 μm. (**B**) Mean intensity fluorescence of autofluorescence of lipofuscin granules. (**C**) Fluorescence microscopy of OP50, US-7, and IT-3-fed-SOD-3::GFP nematode strain at the stage of 2-day adult. Scale bar = 100 μm. (**D**) Mean intensity fluorescence of *C. elegans* SOD-3::GFP transgenic strain fed OP50, US-7, or IT-3. (**E**) Measurement of ROS levels in N2 worms fed US-7 and IT-3 at 2 days of adulthood, compared to OP50 fed nematodes. Statistical analysis was evaluated by one-way ANOVA with the Bonferroni post-test; asterisks indicate significant differences (* *p* < 0.05, ** *p* < 0.01; *** *p* < 0.001). Bars represent the mean of three independent experiments.

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
