# Peer review of "A Combined Proteomics, Metabolomics and In Vivo Analysis Approach for the Characterization of Probiotics in Large-Scale Production"

_biomolecules, 2020, doi:10.3390/biom10010157_

Round 1

Reviewer 1 Report

In this paper, the authors used a combination of proteomics, metabolomics and in vivo studies to compare VSL #3 from two different production sites. Though the analyses are extensive, it is not clear what advances this study achieves compared with previous studies, as a few studies already revealed functional differences between the US VSL #3 and Italy VSL #3 (Cinque et al., 2016 & 2017; Palumbo et al., 2019), and it is not surprising to find any differences in proteomics and metabolomics. I would like to hear how the authors address the following questions:

What insights can we gain from the proteomic and metabolomic results? As the authors mentioned in the discussion, it is not clear whether those differences are attributed to fluctuations in relative concentration of individual strains, or to their metabolic activity. If it is purely dependent on the composition of the strains, why would we bother to conduct the proteomic and metabolomic experiments? From this study, can we gain more meaningful knowledge about how manufacturing procedures affect metabolic activity of the probiotics than "a generalized abundance shift of metabolic proteins"?

Given that a few studies already revealed functional differences between the US VSL #3 and Italy VSL #3 (Cinque et al., 2016 & 2017; Palumbo et al., 2019), I recommend the authors to stress why this study is different from previous ones and point out the advantages of using C. elegans as a model organism in the introduction section.

For the in vivo experiments, some controls are lacking. For instance, OP50 control is required in Figure 6B, C &D.

Minor points:

Quantification is required for Figure 6A. It would be helpful to include the lifespan results as a supplemental figure. Methods on the C. elegans part are too brief. For example, how many worms were used for each assay? If not described in the methods, they should be indicated in the figure legends. What microscopes and filters were used to examine the fluorescence? Typos and mistakes:

Line 80-81: There is a grammar mistake in “Batches were maintained according manufacturer’s until use.”

Line 383-384: Only Figure 6C shows fluorescence intensity of SOD-3::GFP

Line 387-388: There is no Figure 6E in Figure 6.

The name “IT-1” was mistakenly used in Section 3.4, whereas it should be “IT-3”.

Lacking references: Cinque et al., 2016 & 2017;

Author Response

We are grateful to the Reviewers for their valuable comments.

All the suggestions were taken into account carefully and the corresponding modifications are highlighted in yellow the text and using the "Track Changes" function in Microsoft Word. The related explanations are hereafter detailed.

Reviewer #1

In this paper, the authors used a combination of proteomics, metabolomics and in vivo studies to compare VSL #3 from two different production sites. Though the analyses are extensive, it is not clear what advances this study achieves compared with previous studies, as a few studies already revealed functional differences between the US VSL #3 and Italy VSL #3 (Cinque et al., 2016 & 2017; Palumbo et al., 2019), and it is not surprising to find any differences in proteomics and metabolomics. I would like to hear how the authors address the following questions:

What insights can we gain from the proteomic and metabolomic results?

We apology for lack of clarity. In this manuscript, our main intent was to demonstrate that proteomics and metabolomics are powerful tools to investigate possible differences occurring among different lots produced in the same or in different facilities. The resulting picture of differential proteins and metabolites could also give insight on shifts in metabolic pathways.

As the authors mentioned in the discussion, it is not clear whether those differences are attributed to fluctuations in relative concentration of individual strains, or to their metabolic activity. If it is purely dependent on the composition of the strains, why would we bother to conduct the proteomic and metabolomic experiments?

All the differences we highlighted, according to extensive functional proteomics and metabolomics analyses, suggested, as mentioned by the reviewer, a possible different biological and metabolic functionality of the investigated products. Despite these latter are reported sharing the same formulation, detected differences may be related to variations in relative abundance of individual strains and/or to their different metabolic activity. The profiles delineated by proteomics analyses indeed represent a valuable source of data to infer functional differences occurring, at a biomolecular level, among the investigated lots. Furthermore, athanks to species-specificity of the protein sequence, protein expression differences can also be assigned to individual strains. Then, the metabolomics analyses were crucial to detect differences between the final products.

In conclusion, fluctuations in protein and metabolite concentration are indicative of functional differences of the biological system under analysis as well as of possible differences in manufacturing performances.  

Obtained data prove the reliability in using a combination of proteomics and metabolomics techniques for quality assurance in probiotic large-scale production. That was the main goal of the manuscript and it is why we decided to conduct proteomics and metabolomics experiments.

The authors sincerely regret for not making clearer this concept and, in order to improve the quality of the manuscript, according to the criticism of the Reviewer #1, they rewrote the majority of the Introduction and modified the title (see also response to Reviewer #2). We hope that the changes introduced may clarify the doubts of the Reviewer #1.  

From this study, can we gain more meaningful knowledge about how manufacturing procedures affect metabolic activity of the probiotics than "a generalized abundance shift of metabolic proteins"?

 Given that a few studies already revealed functional differences between the US VSL #3 and Italy VSL #3 (Cinque et al., 2016 & 2017; Palumbo et al., 2019), I recommend the authors to stress why this study is different from previous ones and point out the advantages of using C. elegans as a model organism in the introduction section.

For the in vivo experiments, some controls are lacking. For instance, OP50 control is required in Figure 6B, C &D.

The experiments with OP50 as control were performed; however, we thought to include only the comparison of the two products. In the revised version the results with OP50 were included.

Quantification is required for Figure 6A. It would be helpful to include the lifespan results as a supplemental figure.

The quantification of Fig.6A has been added in the revised version of the figure and the lifespan data has been added in the supplemental Table S10.

 Methods on the C. elegans part are too brief. For example, how many worms were used for each assay? If not described in the methods, they should be indicated in the figure legends. What microscopes and filters were used to examine the fluorescence?

We apologize for the lack of clarity; the methods have been modified as required.

 Typos and mistakes:

Line 80-81: There is a grammar mistake in “Batches were maintained according manufacturer’s until use.”

Line 383-384: Only Figure 6C shows fluorescence intensity of SOD-3::GFP

Line 387-388: There is no Figure 6E in Figure 6.

The name “IT-1” was mistakenly used in Section 3.4, whereas it should be “IT-3”.

Lacking references: Cinque et al., 2016 & 2017;

All the typos and mistakes have been corrected, accordingly.

Reviewer 2 Report

- The title is not completely clear about the gap of knowledge you cover with this article. Could you please improve it?

- Did you have any data able to provide information about the cell count of the different species/strains in the analyzed products? It would be important considering that some of your findings could be also addressable to different cell concentrations in the diverse probiotic formulations.

- Please improve the keywords (e.g. using all the possible keywords the journal allowed). Please change in ‘proteomics’.

- Line 23: health-promoting

- Line 41: depending on

- Lines 42-44: “Foligne et al. proved that two different strains of Bifidobacterium animalis subsp. lactis, BL04 and BI07, had de facto different immune profiling and distinct protective effects in a mouse model of colitis [3].” In my opinion, this part does not well fit in this context. You are not introducing strain-dependent characters within a given species, but manufacture-dependent characters of given strains. I would remove the sentence.

- Lines 44-46: Could you please better explain the experimental modes/variables studied in the articles [4-6]. I was expected to find a clear detail about the gap of knowledge that remains unexplored in the studies [4-6]. On this basis, I was expected to find a brief description of the experimental plan you conceived to cover this gap of knowledge.

- Line 53: the environment

- Section 2.1: please report the extended names for all bacterial species. Please do the same for C. elegans.

- Line 246: please evaluate to delete ‘in’

- Line 252: US-prods were?

- Line 318: abundance of differing proteins.

- Line 318: Besides?

- Line 360: utilized as a model organism to screen for

- Line 386: non-fluorescent

- Line 439: health-promoting

- Line 441: pro-longevity

- Line 459: environmental

- Could you please improve the connection between the molecular investigation and the in vivo analysis?

- Could you please improve the discussion concerning the biological phenomena behind the molecular differences you highlighted? E.g. you did not mention the stress response phenomena https://www.tandfonline.com/doi/abs/10.1080/10408398.2019.1580673

Author Response

We are grateful to the Reviewers for their valuable comments.

All the suggestions were taken into account carefully and the corresponding modifications are highlighted in yellow the text and using the "Track Changes" function in Microsoft Word. The related explanations are hereafter detailed.

Reviewer #2

The title is not completely clear about the gap of knowledge you cover with this article. Could you please improve it?

The title has been modified as follow: “A combined proteomics, metabolomics and in vivo analysis approach for quality assessment in large-scale production of probiotics”

- Did you have any data able to provide information about the cell count of the different species/strains in the analyzed products? It would be important considering that some of your findings could be also addressable to different cell concentrations in the diverse probiotic formulations.

We agree with the Reviewer #2 concern, however, we have no data about cell count of the final products. We think that in our experimental conditions the cell count could be misleading since it does not take into account cell debris that can be present in variable amount in final products and that could contribute to our analysis. The aim of our work is to validate a multidisciplinary approach for probiotic quality assessment and, for this reason, we decided to analyze the products in toto.

- Please improve the keywords (e.g. using all the possible keywords the journal allowed). Please change in ‘proteomics’.

In agreement with the Reviewer #2 indications the keywords have been modified as follow:

probiotic quality assessment; Caenorhabditis elegans; functional proteomics; metabolomics; oxidative stress; aging.

- Line 23: health-promoting

- Line 41: depending on

The words have been corrected.

- Lines 42-44: “Foligne et al. proved that two different strains of Bifidobacterium animalis subsp. lactis, BL04 and BI07, had de facto different immune profiling and distinct protective effects in a mouse model of colitis [3].” In my opinion, this part does not well fit in this context. You are not introducing strain-dependent characters within a given species, but manufacture-dependent characters of given strains. I would remove the sentence.

As suggested by the Reviewer #2, the sentence has been removed.

- Lines 44-46: Could you please better explain the experimental modes/variables studied in the articles [4-6]. I was expected to find a clear detail about the gap of knowledge that remains unexplored in the studies [4-6]. On this basis, I was expected to find a brief description of the experimental plan you conceived to cover this gap of knowledge.

We thank the Reviewer #2 for the comment and we apologize for the lack of clarity. The aim of our work is to understand how differences at molecular level (proteomics and metabolomics) of the two formulates could be related to the different proprieties characterized by previous works [4-6] and to validate a combined approach useful in assessing the quality of probiotics on large scale production. In the revised version, some explanatory sentences were thus added in the introduction to clarify the experimental plan.

- Line 53: the environment

- Section 2.1: please report the extended names for all bacterial species. Please do the same for C. elegans.

- Line 246: please evaluate to delete ‘in’

- Line 252: US-prods were?

- Line 318: abundance of differing proteins.

- Line 318: Besides?

- Line 360: utilized as a model organism to screen for

- Line 386: non-fluorescent

- Line 439: health-promoting

- Line 441: pro-longevity

- Line 459: environmental

All the words have been modified as indicated

- Could you please improve the connection between the molecular investigation and the in vivo analysis?

We thank the Reviewer #2 for his/her comment that helped to improve the manuscript. Sentences have been added in the Discussion of the revised manuscript to strength the relation between the molecular and in vivo analysis.

- Could you please improve the discussion concerning the biological phenomena behind the molecular differences you highlighted? E.g. you did not mention the stress response phenomena https://www.tandfonline.com/doi/abs/10.1080/10408398.2019.1580673

We thank the Reviewer #2 for the precious suggestion. In agreement with his/her concerns, a consistent digression has been added in the Discussion and the reference [Fiocco et al., 2019] has been included in the revised manuscript.

Round 2

Reviewer 1 Report

I appreciate the authors’ efforts in revising the manuscript, and the rewriting of the introduction and the discussion indeed has addressed some of my questions. However, I think a more concise introduction and discussion would be easier to read. I especially had a difficult time trying to figure out the point of each paragraph added into the discussion. I would recommend the authors to properly abridge both the introduction and the discussion.

I do not agree with the change of the title from “characterization” to “quality assessment”. Unless the paper defines “quality”, or sets a standard for it, or makes a conclusion about which product has a better quality than the other, I think using “quality assessment” is not appropriate.

Grammar errors:

Line 506 – 507: to which the probiotic strains may have been underwent

Line 524: they allowed to hypothesized

Author Response

We thank the Reviewer for the comments.

I appreciate the authors’ efforts in revising the manuscript, and the rewriting of the introduction and the discussion indeed has addressed some of my questions. However, I think a more concise introduction and discussion would be easier to read. I especially had a difficult time trying to figure out the point of each paragraph added into the discussion. I would recommend the authors to properly abridge both the introduction and the discussion.

We apologize for the lack of clarity, the introduction and discussion have been modified as suggested.

 I do not agree with the change of the title from “characterization” to “quality assessment”. Unless the paper defines “quality”, or sets a standard for it, or makes a conclusion about which product has a better quality than the other, I think using “quality assessment” is not appropriate.

In agreement with the review concern, we change the title with “A combined proteomics, metabolomics and in vivo analysis approach for the characterization of probiotics in a large-scale production.”

Grammar errors:

Line 506 – 507: to which the probiotic strains may have been underwent

Line 524: they allowed to hypothesized.

The text has been modified as required.

Reviewer 2 Report

In the whole manuscript, please italicized the terms 'in  vitro' and 'in vivo'.

Author Response

We thank the reviewer, the text has been corrected accordingly.